# Spatial expression analyses of the putative oncogene ciRS-7 in cancer reshape the microRNA sponge theory

Lasse S. Kristensen [1,2,3✉], Karoline K. Ebbesen[1,2], Martin Sokol [4], Theresa Jakobsen[1], Ulrik Korsgaard[4], Ann C. Eriksen[4], Thomas B. Hansen [1,2], Jørgen Kjems[1,2] & Henrik Hager[4,5✉]

Circular RNAs (circRNAs) have recently gained substantial attention in the cancer research field where most, including the putative oncogene ciRS-7 (CDR1as), have been proposed to function as competitive endogenous RNAs (ceRNAs) by sponging specific microRNAs. Here, we report the first spatially resolved cellular expression patterns of ciRS-7 in colon cancer and show that ciRS-7 is completely absent in the cancer cells, but highly expressed in stromal cells within the tumor microenvironment. Additionally, our data suggest that this generally apply to classical oncogene-driven adenocarcinomas, but not to other cancers, including malignant melanoma. Moreover, we find that correlations between circRNA and mRNA expression, which are commonly interpreted as evidence of a ceRNA function, can be explained by different cancer-to-stromal cell ratios among the studied tumor specimens. Together, these results have wide implications for future circRNA studies and highlight the importance of spatially resolving expression patterns of circRNAs proposed to function as ceRNAs.

[1] Department of Molecular Biology and Genetics (MBG), Aarhus University, C.F. Møllers Allé 3, DK-8000 Aarhus, Denmark. [2] Interdisciplinary Nanoscience Center (iNANO), Aarhus University, Gustav Wieds Vej 14, DK-8000 Aarhus, Denmark. [3] Department of Biomedicine, Aarhus University, Høegh-Guldbergs Gade 10, DK-8000 Aarhus, Denmark. [4] Department of Clinical Pathology, Vejle Hospital, Beriderbakken 4, DK-7100 Vejle, Denmark. [5] Danish Colorectal Cancer Center South, Vejle Hospital, Beriderbakken 4, DK-7100 Vejle, Denmark. ✉email: lasse@biomed.au.dk; Henrik.Hager@rsyd.dk

In recent years, circular RNAs (circRNAs) have emerged as a large class of primarily non-coding endogenous transcripts generated by an alternative splicing event, which covalently links a splice-donor site to an upstream splice-acceptor site[1–3]. The circRNAs are mainly located in the cytoplasm where they have diverse functions related to the binding of other molecules, including microRNAs (miRNAs) and proteins[4]. In particular, our discovery of a circRNA named ciRS-7 (also known as CDR1as), which harbors 63 highly conserved binding sites for miR-7[2,5], stimulated the idea that circRNAs, in general, may function as competitive endogenous RNAs (ceRNAs)[6] by sponging miRNA molecules and thereby relieving the corresponding miRNA target genes from post-transcriptional repression. Since then, there has been a strong interest in discovering differentially expressed cir-cRNAs in cancer and most of these circRNAs were proposed to function as ceRNAs[7].

Shortly after its discovery, ciRS-7 was proposed to have oncogenic properties[8], since miR-7 is known as a tumor sup-pressor that targets and downregulates central oncogenic factors such as EGFR[9], IGF1R[10], and PIK3CD[11]. An oncogenic function of ciRS-7 was reinforced by several studies observing that this circRNA is overexpressed in tumors relative to adjacent normal- or healthy control tissues[12–23]. In colorectal cancer, high expression of ciRS-7 was shown to correlate with poor prognosis, and overexpression of ciRS-7 in colon cancer cell lines showed that it may exert its oncogenic function by sequestering miR-7, leading to activation of the miR-7 target genes EGFR and RAF1[18].

Strikingly, the spatial expression patterns of circRNAs at single-cell level are rarely investigated in cancer and have not been explored for ciRS-7. Therefore, the identity of ciRS-7 expressing cells within tumors and the degree of intra-tumor heterogeneity with respect to ciRS-7 expression has not been explored.

Here, we show that colon cancer cells completely lack ciRS-7 expression in vivo, but find that ciRS-7 is abundantly expressed within tumor stromal cells. Moreover, by correlating the expression of 20 miR-7 target genes with ciRS-7 expression in colon cancer tissue samples, we find that ciRS-7 correlates posi-tively with stromal cell-enriched genes and negatively with cancer cell-enriched genes. We observe similar correlations with miR-7 target genes for other stromal cell-enriched circRNAs, which do not harbor miR-7 binding sites. These data cannot be explained by the ceRNA hypothesis, but may instead be related to differ-ences in cancer-to-stromal cell ratios within the samples. Finally, we find that ciRS-7 expression is generally absent in cancer cells of other classical oncogene-driven adenocarcinomas.

## Results

**ciRS-7 is not expressed in colon cancer cells**. The circular RNA ciRS-7 has been described as a putative oncogene in several cancers including breast cancer[20], esophageal squamous cell carcinoma[14], hepatocellular carcinoma[22], non-small-cell lung cancer[15], and colorectal cancer[18]. However, none of these studies provided in situ data to assess potential intra-tumor heterogeneity related to the expression of ciRS-7. In the present study, we included 32 stage II and III colon cancer patients. The clin-icopathological patient characteristics are shown in Table 1. When applying a recently developed, sensitive, and specific RNA chromogenic in situ hybridization (CISH) assay for spatial visualization of ciRS-7 expression[24] to analyze samples from 16 representative patients, surprisingly, we found that ciRS-7 was not expressed in colon cancer cells in any of the samples. On the other hand, we observed a strong staining in stromal cells within the tumors (Fig. 1a, b). Moreover, we observed that ciRS-7 is not expressed in normal colon epithelial cells nor in adjacent

**Table 1 Clinicopathological characteristics of the patient cohort.**

| Variable | Distribution |
|---|---|
| *Age (years)* | |
| <65, *n* (%) | 6 (18.8) |
| ≥65, *n* (%) | 26 (81.2) |
| *Sex* | |
| Male, *n* (%) | 17 (53.1) |
| Female, *n* (%) | 15 (46.9) |
| *Primary tumor location* | |
| Cecum, *n* (%) | 7 (21.9) |
| Colon ascendens, *n* (%) | 3 (9.4) |
| Right Colic Flexure, *n* (%) | 2 (6.3) |
| Colon Transversum, *n* (%) | 6 (18.9) |
| Left Colic Flexure, *n* (%) | 0 (0.0) |
| Colon descendens, *n* (%) | 0 (0.0) |
| Colon sigmoideum, *n* (%) | 14 (43.8) |
| *Perineural invasion* | |
| Yes, *n* (%) | 2 (6.3) |
| No, *n* (%) | 21 (65.6) |
| Unknown, *n* (%) | 9 (28.1) |
| *Lymphatic or vascular invasion* | |
| Yes, *n* (%) | 5 (15.6) |
| No, *n* (%) | 22 (68.8) |
| Unknown, *n* (%) | 5 (15.6) |
| *Tumor differentiation* | |
| Well or moderately differentiated, *n* (%) | 27 (84.4) |
| Poorly differentiated or undifferentiated, *n* (%) | 5 (15.6) |
| *Mismatch repair status* | |
| Mismatch repair-deficient, *n* (%) | 6 (18.8) |
| Mismatch repair-proficient, *n* (%) | 26 (81.3) |
| Unexamined, *n* (%) | 0 (0.0) |
| *T stage* | |
| T1, *n* (%) | 0 (0.0) |
| T2, *n* (%) | 0 (0.0) |
| T3, *n* (%) | 27 (84.4) |
| T4, *n* (%) | 5 (15.6) |
| *N stage* | |
| N0, *n* (%) | 27 (84.4) |
| N1, *n* (%) | 5 (15.6) |
| *M stage* | |
| M0, *n* (%) | 32 (100.0) |
| M1, *n* (%) | 0 (0.0) |

adenoma tissue and cancer cells of lymph node metastases, but that adjacent adipocytes do express ciRS-7 (Supplementary Fig. 1). To ascertain that lack of ciRS-7 expression in cancer cells is a general phenomenon, we performed the same in situ analyses for ciRS-7 on an additional 47 samples from colon cancer patients using tissue microarrays (TMAs). Again, we found that the cancer cells do not express ciRS-7, whereas the tumor stromal cells do (Supplementary Fig. 2). Importantly, we confirmed that our CISH assay is specific for ciRS-7 by analyzing ciRS-7 knock-out and wild-type cells (Supplementary Fig. 3). Moreover, to confirm the observed expression patterns of ciRS-7 using another method, we performed laser capture microdissection (LCMD) of cancer and stromal cells from four representative patient samples and ana-lyzed the pools of cells separately using NanoString nCounter technology[25]. This analysis revealed a 45-fold higher expression of ciRS-7 in stromal cells relative to cancer cells. We repeated this experiment using another 4 patient samples and found a 37-fold higher expression of ciRS-7 in stromal cells relative to cancer cells (Fig. 1c). Together, these analyses firmly establish that ciRS-7 is not expressed in colon cancer cells, most likely because it is not expressed in the cancer-initiating epithelial cells.

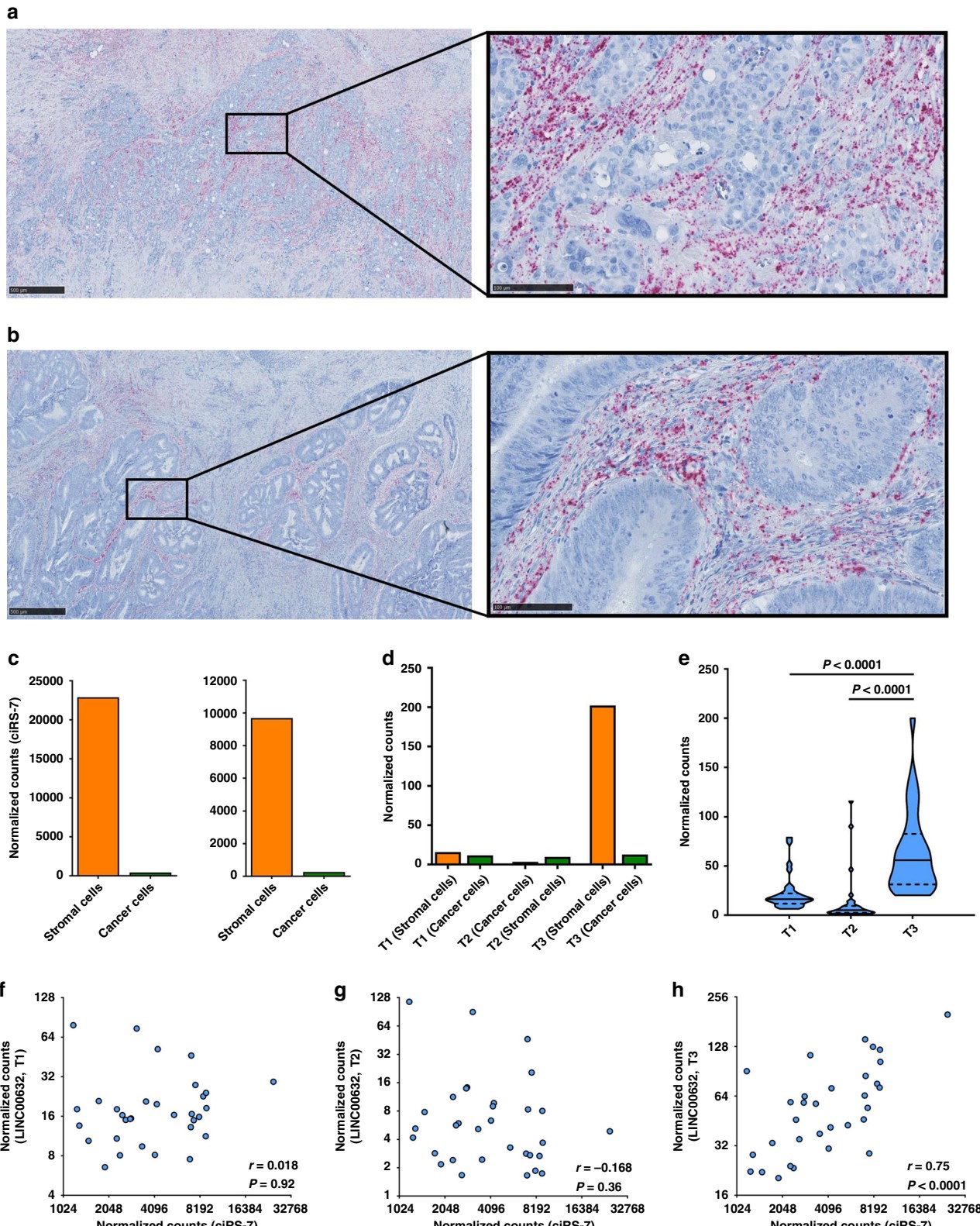

**The T3 transcript of LINC00632 may drive ciRS-7 expression**. It has recently been shown that ciRS-7 is produced from a locus currently annotated as a long non-coding RNA (LINC00632)[26,27], which is located upstream of ciRS-7 and encode three different transcripts denoted as T1, T2, and T3[27]. We assessed the expression levels of these transcripts in the LCMD samples and found that only T3 was expressed and almost exclusively in the stromal cells (Fig. 1d), supporting that the T3 transcript of LINC00632 drives ciRS-7 expression in the stromal cells. Then, we analyzed samples from the 32 colon cancer patients (Table 1) using NanoString nCounter technology and found that T3 was the most abundantly expressed of the three transcripts (Fig. 1e). Moreover, no significant correlations with ciRS-7 expression were observed for T1 and T2 (Fig. 1f, g), whereas the expression of T3

**Fig. 1 The putative oncogene, ciRS-7, is not expressed in colon cancer cells. a, b** RNA chromogenic in situ hybridization (CISH) analyses for ciRS-7 in a representative poorly differentiated (**a**) and a representative well differentiated (**b**) colon adenocarcinoma. Samples from 16 representative colon cancer patients, described in Table 1, were tested and showed similar results. Overviews (left) and higher magnifications (right), indicated in the overview with a square, are shown. The ciRS-7 signal (red dots) is observed in the tumor stroma whereas the cancer cells are negative. Scale bars, corresponding to 500 μm (overviews) and 100 μm (higher magnifications), are indicated in the lower-left corners. **c, d** NanoString nCounter expression analyses of ciRS-7 (two individual experiments each with n = 1) (**c**) and of the transcripts T1, T2 and T3 of LINC00632 (n = 1) (**d**) in laser capture microdissected cancer- and stromal cells pooled from four individual representative patient samples. **e** Violin plots showing expression levels of T1, T2, and T3 in 32 samples from colon cancer patients measured by NanoString nCounter analyses. Quartiles and medians are indicated. Statistical analysis were performed by two-tailed Mann–Whitney test. **f, h** Scatterplots showing the expression levels in 32 colon cancer patients samples of ciRS-7 (x-axis) and T1 (**f**), T2 (**g**), and T3 (**h**) (y-axis) with corresponding linear regression statistics, employing an F test to investigate if the slope was significantly non-zero, and Pearson correlation coefficients (r). Correction for multiple testing was not performed.

correlated positively with the expression of ciRS-7 (r = 0.75, P < 0.0001) (Fig. 1h). The observation that none of the LINC00632 transcripts were expressed in the cancer cells further supports the notion that ciRS-7 is not expressed in these cells, and the selective expression of the T3 transcript in the stromal cells support the notion that ciRS-7 expression is stromal cell-specific.

**Correlations between ciRS-7 and miR-7 target genes**. Numerous studies of ciRS-7 in cancer found correlations between the expression of ciRS-7 and miR-7 target genes[12–22]. Therefore, we decided to study the relationship between ciRS-7 expression and 20 putative miR-7 target genes selected from the literature[8,12–22]. If the classical sponge theory/ceRNA hypothesis[8,18] applies, there should be a positive correlation between expression of ciRS-7 and miR-7 target genes. Notably, upon analysis of bulk RNA in the 32 colon cancer patient samples, we found that ciRS-7 was significantly positively correlated with *FOS* (r = 0.84, P < 0.0001), *NR4A3* (r = 0.80, P < 0.0001) and *PIK3CD* (r = 0.40, P = 0.02). On the other hand, ciRS-7 was significantly negatively correlated with *PAK1* (r = −0.55, P = 0.001) and *CDK1* (r = −0.39, P = 0.03) (Fig. 2a–e). Strikingly, upon analysis of the expression of all 20 putative miR-7 target genes in the pooled fractions of cancer and stromal cells isolated by LCMD (Fig. 2f), we observed that *FOS*, *NR4A3*, and *PIK3CD* presented the highest expression in stromal cells relative to cancer cells being 26, 19, and 12 fold more abundant, respectively. Using immunohistochemistry we confirmed that *FOS* is primarily expressed in the stromal cells (Supplementary Fig. 4). In contrast, *PAK1* and *CDK1* were much more abundant in cancer cells compared to stromal cells. Thus, while the positive correlations between ciRS-7 and putative miR-7 target genes could theoretically be explained by the ceRNA hypothesis, the negative correlations are not compatible with ciRS-7 sequestering miR-7 from its target genes.

To gain further insight into the driver(s) of the observed positive correlations, we used a CRISPR/Cas9 based strategy to remove the ciRS-7 locus and reporter assays in HEK293T cells (Fig. 2g–l). Because *FOS* showed the strongest positive correlation with ciRS-7 when analyzing the tissue samples, we focused on this particular putative miR-7 target. PCR analysis of genomic DNA from the cell population verified that the deletion was only present in cells transfected with ciRS-7 specific sgRNAs (Fig. 2h) and a substantial reduction in ciRS-7 expression was evident (Fig. 2i). Surprisingly, we found no considerable change in *FOS* expression upon ciRS-7 knockout in these populations (Fig. 2j). Moreover, we tested whether *FOS* expression levels responded to altered miR-7 expression using a luciferase reporter setup. Surprisingly, we found no difference in relative luminescence when co-expressing miR-7 or miR-769 as a control (Fig. 2k, l). These results were confirmed using a previously developed dual-color fluorescent reporter (ppC) that allows single-cell measurements[28] (Supplementary Fig. 5). This experiment also showed that the *FOS* reporter did not elicit the degradation of miR-7 (Supplementary

Fig. 5). Taken together, these results argue against explaining the observed correlations using the ceRNA hypothesis. Moreover, we profiled the expression of 799 miRNAs in the laser capture microdissected fractions of cancer and stromal cells, and found that miR-7 was only expressed in the cancer cells (Fig. 2m and Supplementary Data 1). This finding was confirmed by in situ profiling of miR-7 (Supplementary Fig. 6). Again, arguing against explaining the observed correlations by the ceRNA hypothesis.

**A simple model may explain the observed correlations**. An alternative explanation for the observed correlations between ciRS-7 and miR-7 target genes could be differences in the relative compositions of stromal and cancer cells within the tissue samples (Fig. 3). This model implies that positive correlations will be observed between genes co-expressed in the same cells, when cells not expressing either of the genes are present in various amounts across the samples (Fig. 3a). On the other hand, negative correlations are expected when gene expression is mutually exclusive between different cell types within the tumors (Fig. 3b). In support of this model, using NanoString nCounter technology, we observed that stromal cell-enriched genes generally correlated positively with ciRS-7 and that cancer cell-enriched genes generally correlated negatively with ciRS-7 (Fig. 4a). When investigating this systematically, we observed a strong correlation (r = 0.84, P < 0.0001) between the cancer-to-stromal cell expression ratios for each of the 20 miR-7 target genes and their respective correlations with ciRS-7 (Fig. 4b). However, if this model indeed explains the majority of the observed correlations, we would expect to see similar correlations between other stromal cell-enriched circRNAs, which do not harbor miR-7 binding sites, and the miR-7 target genes. Therefore, we analyzed the expression of an additional 11 circRNAs in the fractions of cancer and stromal cells isolated by LCMD. Most of these circRNAs (8 of 11) were more abundant in the stromal cells (Fig. 5a). We selected two stromal cell-enriched circRNAs previously implicated in cancer (circCCDC66[29] and circFBXW7[30,31]) and two cancer cell-enriched circRNAs (circZKSCAN1 and circZNF91) expressed at reasonable levels, none of which harbor miR-7 binding sites according to CircInteractome[32]. As predicted by our model, we observed positive correlations for stromal cell-enriched genes and negative correlations for cancer cell-enriched genes for both circFBXW7 and circCCDC66 (Fig. 5b, c and Supplementary Fig. 7), similar to what we observed for ciRS-7 (Fig. 4). In addition, and as predicted by the model, we found negative correlations with stromal cell-enriched genes and positive correlations with cancer cell-enriched genes for both circZKSCAN1 and circZNF91 (Fig. 5d, e and Supplementary Fig. 7).

**Expression of ciRS-7 in a selection of other cancer types**. Finally, we investigated whether ciRS-7 is expressed in the cancer cells of a selection of other cancers using a TMA. This included

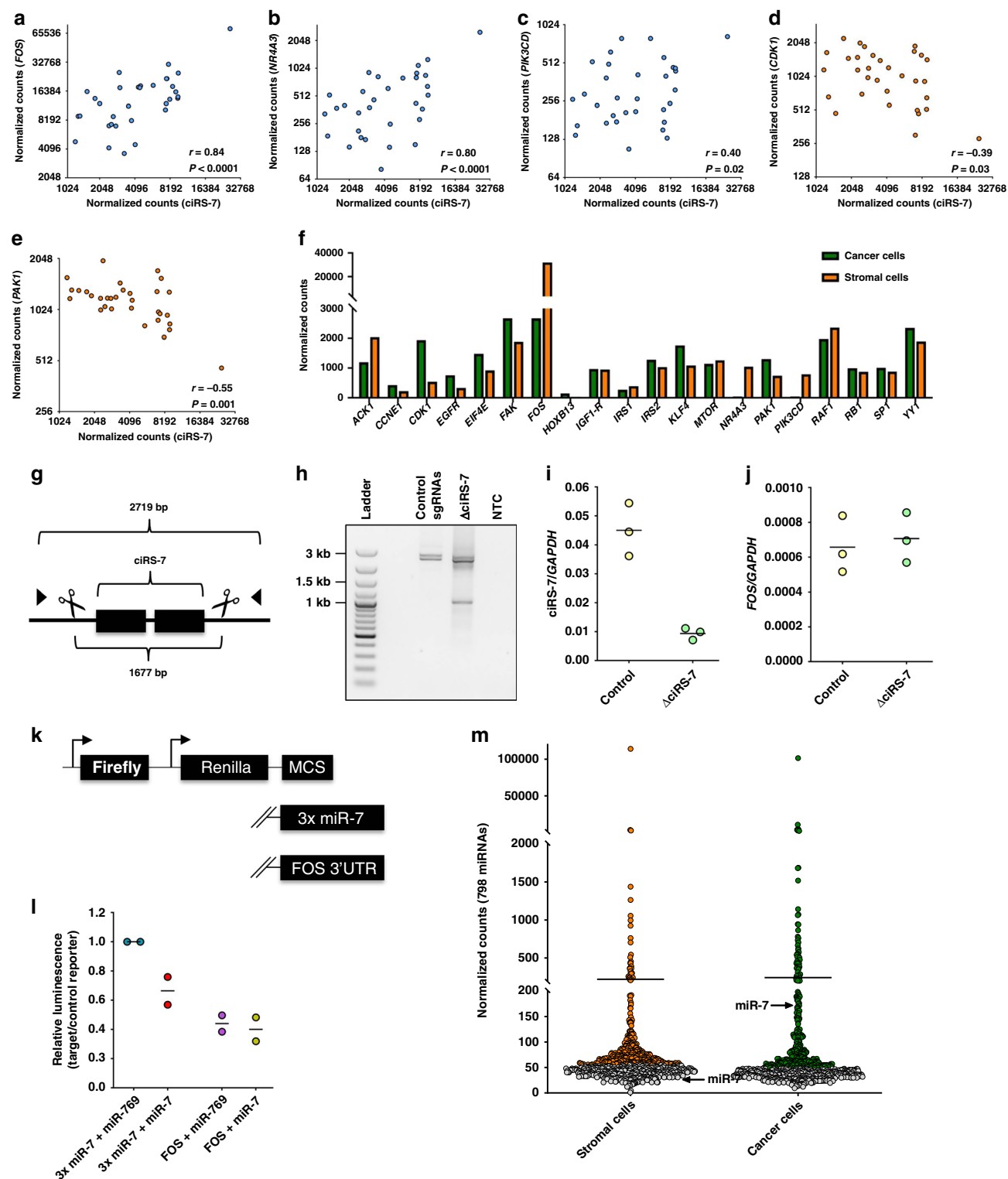

adenocarcinomas of the cervix, lung, breast, ovarium, pancreas, and stomach, none of which showed expression of ciRS-7 in the cancer cells (Fig. 6), as well as embryonal carcinoma, seminoma, medullary thyroid carcinoma, neuroendocrine pancreatic carcinoma, rhabdomyosarcoma, and malignant melanoma, which showed expression of ciRS-7 in the cancer cells (Supplementary Fig. 8). Together these analyses indicate that cancer cells of classical oncogene-driven adenocarcinomas do not express ciRS-7, whereas strong expression can be observed in the stromal cells within these tumors. On the other hand, the cancer cells of germinal- and

neuroendocrine carcinomas express ciRS-7. This also applied to malignant melanoma, which arises from the neural crest-derived melanocytes, and some mesenchymal tumors. These preliminary results indicate that the cellular expression of ciRS-7 in tumor tissues is cancer type-specific and warrants further investigation.

## Discussion
While the circRNA, ciRS-7, has previously been shown to (1) be more abundant in tumors relative to adjacent normal tissues,

**Fig. 2 Correlations between ciRS-7 and miR-7 target genes. a–e** Scatterplot showing expression levels in 32 colon cancer patients samples of ciRS-7 (x-axis) and *FOS* (**a**), *NR4A3* (**b**), *PIK3CD* (**c**), *CDK1* (**d**), and *PAK1* (**e**) (y-axis) with corresponding linear regression statistics, employing an F test to investigate if the slope was significantly non-zero, and Pearson correlation coefficients (*r*). Correction for multiple testing was not performed. **f** NanoString nCounter expression analyses of 20 miR-7 target genes in fractions of stromal and cancer cells isolated by laser capture microdissection of colon cancer tissues, pooled from four individual representative patient samples ($n = 1$). **g** Schematic of approach used to knockout ciRS-7 in HEK293T cells. The expected deletion (1677 bp), the location of PCR primers used to validate deletion of the ciRS-7 exons and the expected amplicon size without the expected deletion (2719) is given. **h** PCR analysis of genomic DNA from HEK293T cells populations transfected with control sgRNAs or sgRNAs for knockout of ciRS-7. **i, j** RT-qPCR analysis of ciRS-7 (**i**) and *FOS* (**j**) expression in the HEK293T cells populations assessed in **h**. The data represent three technical replicates from $n = 1$ biologically independent samples. **k** Schematic of luciferase vector constructs. // indicates that the sequence of the plasmid on the left hand of the schematic is identical to the construct listed at the top. **l** Dual luciferase assay in HEK293T cells. Relative luminescence (Renilla/Firefly)) of target reporter (3x miR-7 or FOS) relative to control reporter (MCS) per replicate is shown ($n = 2$ biologically independent samples). Luciferase reporters were co-transfected with miR-769 (negative control) or miR-7 in a 1:1 ratio. All values are relative to that of the control setup (psicheck 3xmiR-7 + miR-769). **m** NanoString nCounter expression analyses of 798 miRNAs in stromal cells and cancer cells. Each dot represent a miRNA. Gray dots represents miRNAs that were expressed at levels below background. The arrows indicate the expression levels of miR-7.

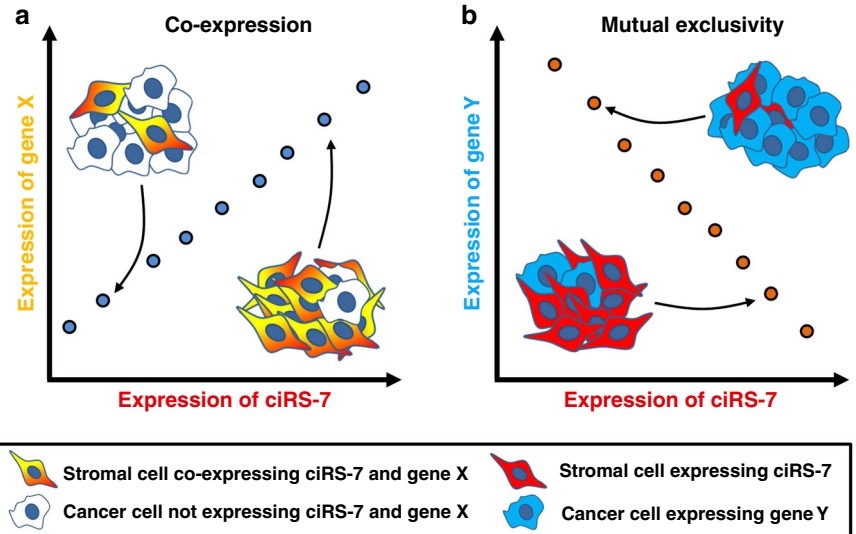

**Fig. 3 A simple model explains correlations between ciRS-7 and miR-7 target genes. a** When ciRS-7 is co-expressed with gene X in the stromal cells, positive correlations will be observed as samples with relatively higher amounts of stromal cells relative to cancer cells will have a relatively higher expression of ciRS-7 and the gene X. **b** When ciRS-7 and gene Y are mutually exclusively expressed in the stromal and cancer cells, negative correlations will be observed as samples with relatively higher amounts of stromal cells relative to cancer cells will have a relatively higher expression of ciRS-7, whereas as samples with relatively higher amounts of cancer cells relative to stromal cells will have a relatively higher expression of the gene Y. Additional cell types with other expression patterns are likely to confound the analyses when analyzing tumor specimens, leading to poorer correlations than illustrated here.

(2) be an indicator of poor prognosis, (3) correlate with the expression of miR-7 target genes in cancer tissues, (4) result in more aggressive oncogenic phenotypes when overexpressed both in vitro and in vivo, we demonstrated that this circRNA is not present in colon cancer cells and in the cancer cells of several other human adenocarcinomas. However, we believe that our data are not in conflict with the majority of the published data, rather it is the interpretations of the data, which are in conflict. Firstly, we did observe a markedly higher expression of ciRS-7 in tumors relative to adjacent normal tissues. While these expression differences have previously been interpreted as the cancer cells overexpressing ciRS-7, we show that they reflect the high abundance of ciRS-7 in stromal cells within tumors relative to uninvolved stromal cells. Secondly, high tumor-stroma ratios have been reported as strong independent prognostic factors in several different adenocarcinomas, including colon[33,34], lung[35], and breast[36,37]. Thus, the high ciRS-7 expression in stromal cells may explain why previous studies found that high ciRS-7 expression correlates with poor prognosis[14,15,18,19,38]. Thirdly, we have demonstrated that the expression of circRNAs correlate with the expression of miR-7 target genes independent of whether the

circRNAs harbor miR-7 binding sites. Instead, circRNAs, including ciRS-7, correlate with the expression of miR-7 target genes depending on whether they are co-expressed or mutually exclusive in the cancer and stromal cells. To explain these findings, we proposed a model based on differences in cancer-to-stromal cell ratios among the tumor specimens. This model elegantly explains why we observed both positive and negative correlations between ciRS-7 and miR-7 target genes and why similar correlations were observed for other stromal cell-enriched circRNAs lacking miR-7 binding sites. Moreover, for cancer cell-enriched circRNAs, we observed the opposite correlations in agreement with the model. We do not observe perfect correlations between cancer-to-stromal cell ratios and correlation of individual circRNAs with the mRNAs (Figs. 4 and 5), likely reflecting the simplicity of the model; the tumor stroma contains several different cell types, and adipocytes and normal tissue may also be present in the specimens. Finally, previous reports showing that overexpression of ciRS-7 in colon cancer cell lines and mouse models leads to more aggressive oncogenic phenotypes are most likely true, but not necessarily physiologically relevant given that ciRS-7 is not expressed in the cancer cells

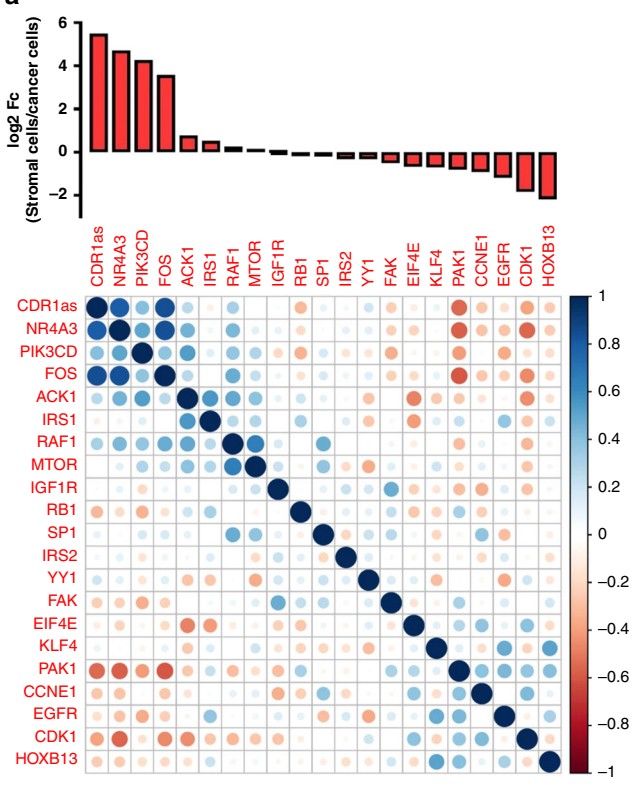

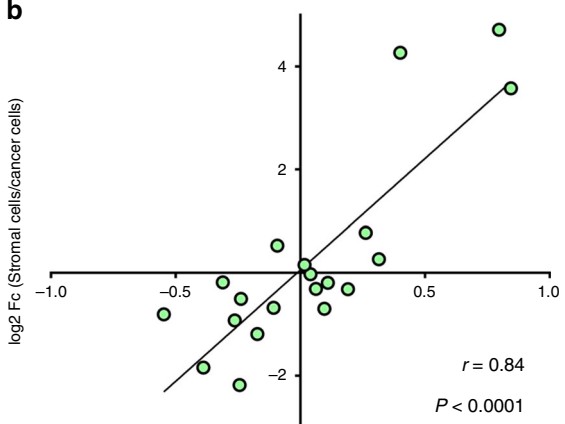

**Fig. 4 ciRS-7 is positively correlated with stromal cell-enriched genes and negatively correlated with cancer cell-enriched genes. a** Correlation matrix showing Pearson correlation coefficients between ciRS-7 and the 20 miR-7 target genes. The genes are listed according to relative expression levels in cancer and stromal cells, displayed in the top. These data were derived from fractions of stromal and cancer cells isolated by laser capture microdissection of colon cancer tissues, pooled from four individual representative patient samples ($n = 1$). **b** Scatter plot showing log2 fold changes in stromal-to-cancer cell ratios (*y*-axis) and correlation with ciRS-7 (*x*-axis) for each of the 20 miR-7 target genes, with corresponding linear regression statistics, employing an F test to investigate if the slope was significantly non-zero, and Pearson correlation coefficient (*r*).

within colon cancer tissues. In addition, not all studies found increased proliferation rates and metastatic potential when overexpressing ciRS-7 in cancer cell lines[38], and loss of ciRS-7 expression has been shown to promote invasion and metastasis of melanoma cells[26]. Finally, while a previous study found a positive

correlation between ciRS-7 and *EGFR* expression[18], we observed a negative correlation between the two.

Although the spatial cellular expression patterns of ciRS-7 have not previously been studied in colon cancer, it has recently been shown that ciRS-7 expression correlates positively with a stromal cell-specific gene expression signature across many different cancer types[39]. These data support our finding that ciRS-7 is not expressed in cancer cells but locates in tumor stromal cells.

The current study is focused on ciRS-7, but our findings are likely to have implications for other ceRNAs in cancer as intra-tumor spatial expression patterns of the proposed ceRNAs and the cognate miRNA target genes are rarely investigated. In particular, a large number of articles suggest that particular circRNAs function as ceRNAs in cancer despite the fact that circRNAs, as a class, do not contain more miRNA-binding sites than would be expected by chance[40]. The vast majority of these studies did not investigate the spatial relationship between the putative circular ceRNA and the potentially post-transcriptionally regulated genes. Moreover, circRNAs are often less abundant in rapidly pro-liferating cells[24,41,42] possibly due to their slow production rate[43] and extreme stability[1] preventing them from reaching steady state levels. This likely results in major expression differences among cell populations with different proliferation rates within a tumor. Therefore, we find it plausible that many published correlations between circRNAs and miRNA target genes may actually be explained by our model (Fig. 3) rather than by the circRNAs functioning as ceRNAs. In support of this notion, modulating the expression of a single endogenous gene will generally not lead to significant changes in competing miRNA-binding sites relative to all of the corresponding miRNA-binding sites present in all mRNAs[44]. Moreover, endogenous stoichiometric relations between the miRNA-binding sites of a proposed ceRNA and the corresponding mRNA target sites of the miRNA are rarely mir-rored in overexpression systems, which therefore often lack physiological relevance[45]. It is also in line with our findings that a recent ciRS-7 knock-out (KO) mouse model suggests that ciRS-7 does not inhibit or facilitate degradation of miR-7. In fact, removal of ciRS-7 resulted in downregulation of miR-7 in neural tissues[46] contrary to what has been observed in studies of bulk RNA from cancer specimens. In contrast, miR-671 was upregu-lated in the KO mouse, consistent with the nearly perfect sequence complementarity with ciRS-7 leads to its degradation[46] and may cause slicing of ciRS-7[47]. Indeed, target RNA-directed miRNA degradation (TDMD) requires extensive com-plementarity outside of the seed sequence[48,49] as exemplified by the long non-coding RNA, *Cyrano*, which induce destruction of miR-7 through a single highly conserved binding site of unusually high complementarity to miR-7[50]. However, extensive com-plementarity outside the seed sequence is rarely observed and often not investigated in studies of circRNA-miRNA interactions in cancer[7].

In our expression data from colon cancer specimens, ciRS-7 correlated most strongly with *FOS*. However, we found no change in *FOS* expression upon ciRS-7 KO in HEK293T cells. We further found that *FOS* failed to respond to miR-7 overexpression using two different reporter assays. Moreover, miR-7 was not expressed in the ciRS-7 expressing stromal cells. Based on these findings, the observed positive correlation between ciRS-7 and *FOS* expression is unlikely due to ciRS-7 mediated regulation of miR-7. The lack of response in the reporter assay may reflect that the human *FOS* 3' UTR, in contrast to the mouse version of *FOS*, does not contain 7-mer or 8-mer target sites for miR-7 according to TargetScan[51], despite previously being described as a miR-7 target gene in humans[52]. Therefore, it should be emphasized that our findings do not exclude that ciRS-7 may function as a regulator of miR-7 and miR-7 targets, as shown in other tissues such as the brain[46,50]

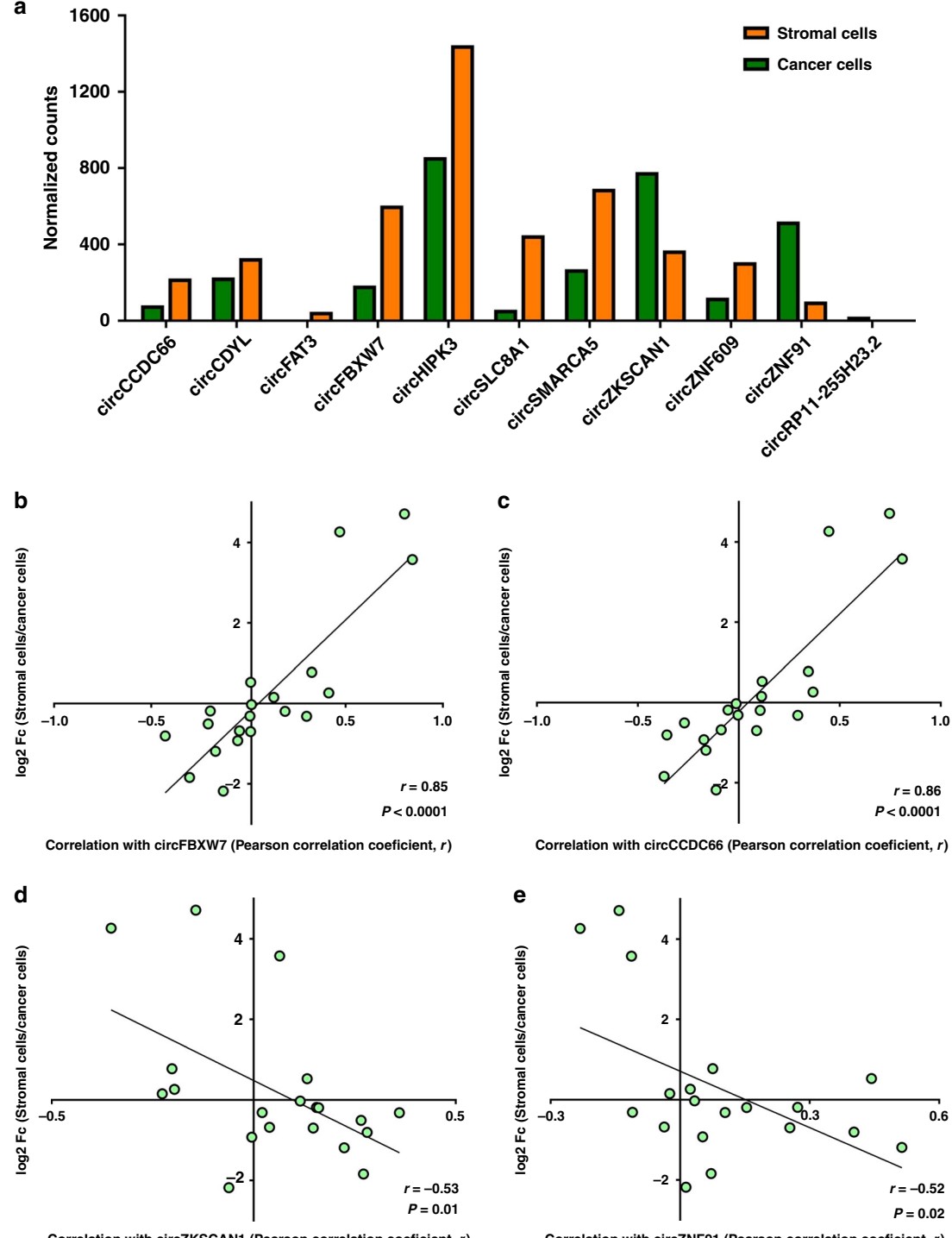

**Fig. 5 Other circRNAs without miR-7 binding sites also correlate with miR-7 target genes. a** NanoString nCounter expression analyses of 11 circRNAs in fractions of stromal and cancer cells isolated by laser capture microdissection of colon cancer tissues, pooled from four individual representative patient samples ($n = 1$). **b, c** Scatter plots showing log2 fold changes in stromal-to-cancer cell ratios (y-axis) and correlation with the stromal cell-enriched circRNAs, circFBXW7 (**b**) and circCCDC66 (**c**) (x-axis), for each of the 20 miR-7 target genes with corresponding linear regression statistics and Pearson correlation coefficient (r). **d, e** Scatter plots showing log2 fold changes in stromal-to-cancer cell ratios (y-axis) and correlation with the cancer cell-enriched circRNAs, circZKSCAN1 (**d**) and circZNF91 (**e**) (x-axis), for each of the 20 miR-7 target genes, with corresponding linear regression statistics, employing an F test to investigate if the slope was significantly non-zero, and Pearson correlation coefficient (r). Correction for multiple testing was not performed.

where it is highly expressed and co-localizes with miR-7[5]. Moreover, our data do not imply that other circRNAs cannot have miRNA-sponging function in cancer. However, differences in cancer-to-stromal cell ratios among tumor specimens can confound the analyses or even result in correlations that may be wrongly interpreted as evidence of ceRNA functions.

Although our data show that ciRS-7 is not expressed in colon cancer cells, further investigations are needed to understand the

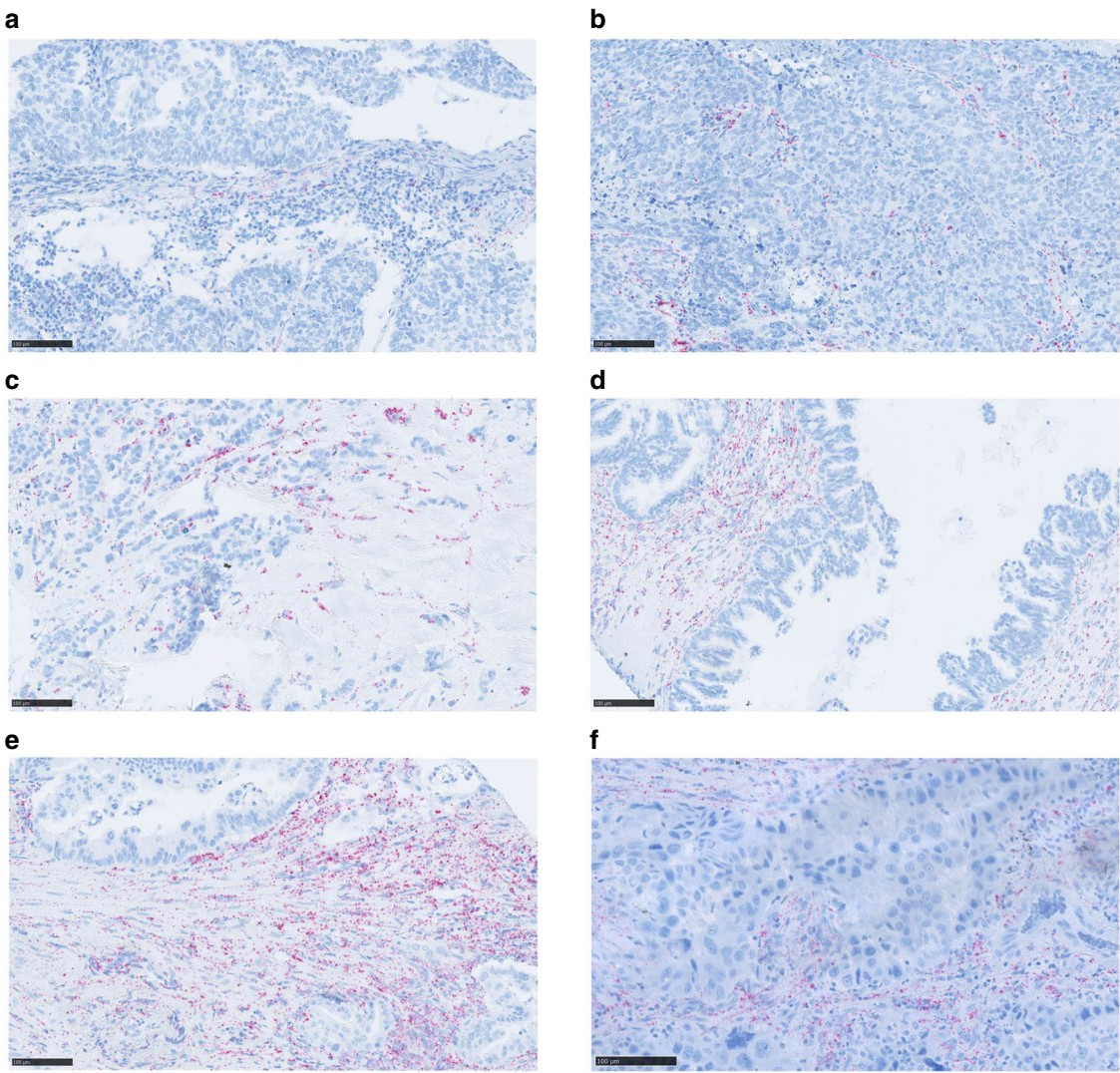

**Fig. 6 RNA chromogenic in situ hybridization (CISH) for ciRS-7 in various adenocarcinomas. a–f** The adenocarcinomas are derived from a tissue microarray and are originating from cervix (**a**), lung (**b**), breast (**c**), ovarium (**d**), pancreas (**e**), and stomach (**f**). For each cancer type at least two cores containing both cancer cells and tumor stroma were analyzed and showed similar results. The ciRS-7 signal (red dots) is observed in the tumor stroma whereas the cancer cells are negative. Scale bars, corresponding to 100 µm, are indicated in the lower-left corners.

role for ciRS-7 in cancer development and progression. The tumor microenvironment (TME) is important in the pathobiology of cancer as stromal cells of the TME restrict the accumulation of T cells near cancer cells[53,54], and ciRS-7 has recently been shown to function independent of miR-7 in melanoma[26]. Indeed, a recent bioinformatics study of ciRS-7 using publicly available RNA-seq datasets from 868 cancer samples strongly suggests that ciRS-7 plays specific roles in immune- and stromal cell infiltration in tumor tissues[39]. Therefore, future research should aim at addressing the potential roles of ciRS-7 in the TME.

In conclusion, our data show that the putative oncogene ciRS-7 is not expressed in colon cancer cells and suggest that this generally apply to classical oncogene-driven adenocarcinomas, whereas ciRS-7 expression in cancer cells can be observed in other cancers, including germinal- and neuroendocrine carcinomas and sarcomas. The previously observed overexpression of ciRS-7 in adenocarcinomas relative to adjacent normal tissues is likely explained by high expression of ciRS-7 in the tumor stromal cells. In addition, we observed that correlations between circRNAs, which are differentially expressed in stromal cells relative to

cancer cells, and miR-7 target genes depend on whether they are co-expressed or mutually exclusive rather than on miR-7 binding sites within the circRNAs. Thus, our findings may have wide implications for studies of ceRNA functions of ciRS-7 and other circRNAs in cancer.

## Methods

**Patient samples**. All patients were treated surgically for stage II or III colon cancer in 2002 in Denmark and identified by a search in the nationwide registry administered by the Danish Colorectal Cancer Group (DCCG). The database contains prospectively collected surgical and pathological data. Tumors located from the cecum to the transversum were defined as right-sided, and those located from the left colonic flexure to the sigmoideum were defined as left-sided. Patient characteristics are summarized in Table 1. In addition, we used TMAs constructed from anonymized formalin-fixed paraffin embedded (FFPE) tissue blocks using the Manual Tissue Arrayer MTA-1. The TMAs included 47 samples from colon cancer patients as well as samples from patients with adenocarcinomas originating from cervix, lung, breast, ovarium, pancreas, and stomach and from patients with embryonal carcinoma, seminoma, medullary thyroid carcinoma, neuroendocrine pancreatic carcinoma, rhabdomyosarcoma and malignant melanoma. For each cancer type at least two cores (1 mm in diameter) containing both cancer cells and tumor stroma were included.

**Ethics**. The study is based on excess surgical specimens from patients with colon cancer and was conducted according to the Declaration of Helsinki Principles. Informed consent from all participants was obtained and the study has been approved by the Regional Ethical Committee of Southern Denmark according to the Danish law (S-20170197 CSF).

**ciRS-7 chromogenic in situ hybridization**. The cellular localization of ciRS-7 was investigated by chromogenic in situ hybridization (CISH) using a modified version of the RNAScope 2.5 high-definition procedure (Advanced Cell Diagnostics [ACD], Hayward, CA, USA)[24]. In brief, 3.5-μm thick paraffin sections from colon cancer samples, TMAs and formalin-fixed and paraffin-embedded cell pellets from ciRS-7 knockout and wild-type cells were pretreated and hybridized with 12 ZZ-pairs (Probe-Hs-CDR1-AS-No-XMm, 510711, ACD) targeting ciRS-7 overnight. The ZZ-pairs binding ciRS-7 were detected using seven amplification steps, including a Tyramid Signal Amplification step (TSA-DIG; NEL748001KT, Perki-nElmer, Skovlunde, Denmark) labeled with alkaline phosphatase-conjugated sheep anti-DIG FAB fragments (11093274910, Roche, Basel, Switzerland), before visualized with Liquid Permanent Red (DAKO, Glostrup, Denmark) and counterstained with Mayer's hematoxylin. Data were collected using a Hamamatsu NanoZoomer XR digital slide scanner and analyzed using NDPview 2 software version 2.7.52 for mac.

**Microdissections**. Five-μm-thick FFPE sections were mounted on membrane slides (Leica, Germany). The sections were deparaffinized for 30 s in Xylene, rehydrated in graded ethanol, stained for 2 s in Mayers hematoxylin and washed in sterile water. After drying, five consecutive sections were microdissected using a microdissection system (LMD 630, Leica Germany). Cancer cells and tumor stromal cells were collected in separate tubes.

**FOS immunohistochemistry**. Immunohistochemistry (IHC) was performed automatically on a Ventana BenchMark Ultra instrument (Ventana Medical System, Tucson, Arizona, USA). The primary antibody used was c-FOS (E8): sc-166940 (Santa Cruz Biotechnology, Dallas, Texas, USA) diluted 1:600. After deparaffination and rehydration of the sections, antigen retrieval was done using EDTA buffer at 100 °C for 30 min. Slides were treated with inhibitors to endogenous peroxidase followed to incubation with primary antibody in 30 min at 37 °C. Amplification was done using Optiview system (Ventana Medical System, Tucson, Arizona, USA). Visualization was done using Diaminobenzidine (DAB) and Hematoxylin was used as counterstain. Human term placenta was used as positive control.

**Cell culture**. HEK293T cells (Invitrogen) were maintained in Dulbecco's modified Eagle's media (DMEM) with GlutaMAX (Thermo Fischer Scientific) supplemented with 10% fetal bovine serum and 1% penicillin/streptomycin sulphate.

**Constructs**. The sgRNAs for CRISPR/Cas9 mediated deletion of ciRS-7 were ordered as oligos, annealed and inserted into the vector pSpCas9(BB)-2A-Puro (PX459) V2.0 (Addgene # 62988) gifted by Feng Zhang.

Two oligos containing three 8mer sites for miR-7 were annealed and inserted into the psiCheck-2 vector (Promega) with a modified multiple cloning site (MCS) using the XhoI/SpeI restriction sites. For the FOS psiCheck-2 reporter the FOS 3′ UTR (800 bp) was PCR amplified and inserted into the multiple cloning site in the 3′ UTR of the Renilla luciferase gene in the same psiCheck-2 vector using the restriction enzymes XhoI and SpeI. The modified psiCheck2-vector without any inserted miRNA target sites was used as a negative control. The vectors expressing miR-7 (pJEBB-miR-7) and miR-769 (pJEBB-miR-769) were described previously[5].

The ppC-dual color plasmid[28] was used as a backbone to generate a ppC-miR-7 reporter containing six 8mer miR-7 target sites. First, the MCS was modified to contain an EcoNI site producing G/C overhang. Then, miR-7 seed containing oligos (miR-7-Seed FW/miR-7-Seed RE) were annealed and phosphorylated, concatemerized, gel purified, and inserted into the EcoNI-digested ppC. The FOS 3′ UTR (800 bp) was PCR amplified and inserted into the MCS in the 3′ UTR of mCherry using the restriction enzymes NotI and XbaI. Here, pcDNA3 was used as the backbone for a miR-7 expression vector (transferred from pJEBB-miR-7 using NotI and XbaI digestion). Primers used to generate these constructs can be found in Supplementary Table 1.

**Knockout of ciRS-7**. For CRISPR/Cas9 mediated removal of ciRS-7, HEK293T cells were transfected 6 h after seeding with vectors encoding sgRNAs and Cas9 using calcium phosphate. Trypsin was used to separate the cells into single cells 24 h after transfection. The cells were then re-plated and transfected cells selected by addition of puromycin (1.5 μg/mL, A11138-03, Invitrogen) for 48 h. The puromycin-containing media was then removed and after washing in 1x PBS, cells were allowed to recover for 24 h before harvesting them for DNA and RNA isolation. Simultaneously, clonal expansion of ciRS-7 knockout and wild-type cells was initiated. Here, the cells were washed twice in 1x PBS, trypsinized, counted and re-plated in 96-well plates at a density of 0.5 cell/well. After two to three weeks, genomic DNA and RNA from the clones was purified and assessed.

Positive ciRS-7 knockout clones and wild-type cells were identified based on PCR used to assess the genomic deletion and absence of ciRS-7 on RNA-level as assessed using RT-qPCR.

**Luciferase reporter assay**. Cells were seeded in 12-well dishes and transfected with 0.1 μg psiCheck-2 vector, 0.1 μg pJEBB-miR-7 or pJEBB-miR-679 in a mix with empty vector (pcDNA3) to equal 1 μg in total. Forty-eight hours after transfection, the cells were harvested and lysed in Passive Lysis Buffer (Promega) according to the manufacturer's instructions. The lysed cells were kept at −20 °C overnight and measured using the dual-luciferase reporter assay kit (Promega) on a BMG FLUOstar luminometer (BMG labtech). The relative luminescence (Renilla/Firefly) was calculated across three replicates and the standard deviation of the mean calculated. For each replicate, values were further normalized to that of the control setup (3xmiR-7 + miR-769).

**Dual-color reporter assay**. HEK293T cells seeded at 50% confluence were transfected using Lipofectamine2000 (Invitrogen) according to the manufacturer's protocol. The transfection mix was composed of a dual-color reporter, a tetra-cycline transactivator (tTA) to allow expression of the dual-color plasmid and either an empty vector (EV) or a miR-7 expression vector (1:1:2 ratio). Forty-eight hours after transfection, cellular GFP and mCherry fluorescent signals were imaged on an Olympus IX73 microscope. eGFP and mCherry signal per cell was quantified in ImageJ version 2.0.0-rc-69/1.52p using a custom macro. In brief, this quantification entailed subtraction of background signal, setting a threshold for signal detection and applying the mask and watershed tools in ImageJ to separate clusters of cells. The eGFP and mCherry signal intensities within each identified particle were measured. Only particles above an empirically set threshold of 200 were included in the analysis.

**RNA and DNA isolation**. Total RNA for each colon cancer sample was isolated from freshly cut 6 × 5 μm whole-tissue sections or from pools of microdissected material using the miRNeasy FFPE Kit (Qiagen, Hilden, Germany) including the DNase I treatment. Commercial deparaffinization solution (Qiagen) was used to remove paraffin from samples prior to RNA isolation. RNA from microdissections was subjected to an optional RNA concentration step using RNA Clean & Concentrator (Zymo Research Europe, Freiburg, Germany). RNA from HEK293T cells was isolated using TRIzol (Thermo Fisher Scientific) according to the manufacturer's instructions. Purified RNA was quantitated on a NanoDrop™ One spectrophotometer (ThermoFisher Scientific) and for the FFPE samples the quality and size distribution of RNA was assessed on a 2200 TapeStation System using High Sensitivity RNA ScreenTape (Agilent, Glostrup, Denmark). Genomic DNA from HEK293T cells was purified using the GenElute™ Mammalian Genomic DNA Miniprep Kit (Sigma-Aldrich) according to the manufacturer's instructions.

**Gene and circRNA expression analyses by NanoString nCounter**. A custom CodeSet of capture- and reporter probes was designed to target 12 circRNAs, including ciRS-7, 20 putative miR-7 target genes, the lncRNA transcripts T1, T2, and T3 variants of LINC000632, as well as seven reference genes (Supplementary Table 2). Approximately 300 ng of FFPE-derived total RNA from each sample was analyzed using the nCounter® FLEX Analysis System (NanoString Technologies, Seattle, WA, USA) analysis according to the manufacturer's instructions using a 24 h hybridization time. The raw data were processed using the nSOLVER 4.0 software (NanoString Technologies); first, a positive control normalization was performed using the geometric mean of all positive controls, then a second normalization using the geometric mean of the four most stable linear reference genes (ACTB, PUM1, MRPL19 and SF3A1) was performed, before exporting the data to Excel 2016 (Microsoft Corporation, Redmond, WA, USA). Data were then plotted using R software version 3.5.1 and Graphpad Prism software version 7.

**Expression analysis using RT-qPCR**. cDNA was synthesized from 0.5 to 1 μg of DNase (Thermo Fisher Scientific) treated and purified RNA using the M-MLV Reverse Transcriptase kit (Thermo Fisher Scientific) according to the manufacturer's instructions using random hexamers. For quantitative analysis of RNA expression, cDNA was mixed with Lightcycler 480 SYBR Green I Master (Roche) and run on a Lightcycler 480 (Roche) using the following program: 95 °C for 10 min, followed by 40 cycles of 95 °C for 10 s, 56 °C for 15 s and 72 °C for 20 s. The experiments were carried out in technical triplicates. The Ct values were obtained using LightCycler 480 SW 1.5.1 software for each triplicate and were transformed ($2^{-Ct}$), averaged (σ) and normalized to the housekeeping gene GAPDH. Primers for qPCR analyses can be found in Supplementary Table 1.

**PCR analysis to confirm ciRS-7 knockout**. PCR was performed on 200 ng DNA using Phusion Hot Start Flex DNA polymerase (New England Biolabs) in GC buffer according to the manufacturer's instructions. The primers used can be found in Supplementary Table 1. The specific program used for PCR consisted of an initial denaturation step (98 °C for 30 s), followed by 35 cycles of denaturation at 98 °C for 10 s, annealing at 63 °C for 15 s and extension step at 72 °C for 2 min and 30 s. Lastly, a final extension step of 72 °C for 5 min was included. For analysis, the

PCR products were run on a 1% agarose gel and assessed based on size and Sanger sequencing of the products.

**miRNA expression analyses by NanoString nCounter**. The nCounter Human v3 miRNA panel (NanoString Technologies), which target 799 miRNAs, was used for miRNA profiling in the fractions of cancer- and stromal cells. One-hundred ng of total RNA from each sample was subjected to nCounter™ SPRINT (NanoString Technologies) analysis according to the manufacturer's protocol using a 20 hour hybridization time. The raw data were processed using the nSOLVER 4.0 software (NanoString Technologies); first, a positive control normalization was performed using the geometric mean of all positive controls, then a second normalization using the geometric mean of the top 100 highest expressed targets was performed, before exporting the data to Excel (Microsoft Corporation, Redmond, WA, USA).

**miR-7 CISH**. The cellular localization of miR-7 was investigated using the miR-NAscope HD Detection Kit (Advanced Cell Diagnostics [ACD], Hayward, CA, USA), according to the manufacturer's protocol. In brief, 3.5-μm-thick paraffin sections were pretreated and hybridized with ZZ-probe (Probe-SR-has miR7-5pS1, ACD) targeting miR-7 for 2 h. The ZZ-pairs binding miR-7 were detected using six amplification steps before visualized with Fat Red and counterstained with Mayer's hematoxylin.

**Expression analyses of miR-7 using PAGE northern blot**. Thirty microgram RNA from HEK293T cells were ethanol precipitated. Then the RNA pellet was dissolved in 40 uL loading buffer (8 M urea, 20 mM EDTA, 1% xylen, 1% bromophenolblue) and loaded onto a 12% PAGE gel that was run for ~1 h and 10 min at 12 W in 1x TBE buffer. The RNA was transferred to a Hybond N + membrane (GE Healthcare) by means of electroblotting overnight at 15 V. The RNA was then UV cross-linked to the membrane and pre-hybridized in Church buffer (0.158 M $NaH_2PO_4$, 0.342 M $Na_2HPO_4$, 7% SDS, 1 mM EDTA, 0.5% BSA, pH 7.5) for one hour. The membrane was probed with a 5′ radioactively labelled DNA oligonucleotide at 37 °C overnight and washed twice in 2x SSC, 0.1% SDS for 10 min at 25 °C before exposure on a phosphoimager screen for data collection on a Typhoon imager. The probe sequences can be found in Supplementary Table 1.

**Statistical analyses**. All statistical tests were performed using Prism 7 (GraphPad, La Jolla, CA, USA). Comparisons of the average expression levels of the T1, T2, and T3 transcripts for LINC00632 in the colon cancer specimens were done using Mann–Whitney test, as the data were not normally distributed according to the D'Agostino & Pearson normality test. Linear regression was used to assess the potential correlation between expression levels of each of the T1, T2 and T3 transcripts for LINC00632 and the expression levels of ciRS-7 in the colon cancer specimens, employing an F test to investigate if the slope was significantly non-zero. The same linear regression analyses were performed to assess potential correlations between expression levels of each of the miR-7 target genes and ciRS-7, as well as circFBXW7, circCCDC66, circZKSCAN1 and circZNF91, in the colon cancer specimens. Likewise, potential correlations between the expression ratios in cancer relative to stromal cells for each of the 20 genes and their respective correlations with ciRS-7 (or circFBXW7, circCCDC66, circZKSCAN1 and circZNF91), were assessed using linear regression. All $P$-values were two-tailed and considered significant if <0.05.

**Reporting summary**. Further information on research design is available in the Nature Research Reporting Summary linked to this article.

## Data availability

Source data are provided with this paper. All other original data that support the findings of this study are available from the corresponding author upon reasonable request.

## Code availability

The computer codes that support the plots within this paper are available from the corresponding author upon reasonable request.

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

## Acknowledgements
We thank biotechnicians Tina Skov, Lone Nielsen and Mariana Semenova for excellent technical assistance. Financial support was granted by the Independent Research Fund Denmark, The Research Council of Lillebaelt Hospital, The Lundbeck Foundation (R307-2018-3433) and The Villum Foundation (00013393).

## Author contributions
L.S.K. and H.H. conceived the study. A.C.E. and H.H. collected the clinical samples used. K.K.E., M.S., and T.J. carried out the laboratory work. H.H. carried out microdissections and H&E stainings. L.S.K., K.K.E., M.S., U.K., T.B.H., and H.H. analyzed the data and prepared the figures and tables. L.S.K., T.B.H., J.K., and H.H. provided reagents and materials. L.S.K. wrote the paper, which was revised for important intellectual content and approved by all authors.

## Competing interests
The authors declare no competing interests.
