## [Peer Review File · Nature Communications]

Reviewers' comments:

Reviewer #1 (Remarks to the Author):

The manuscript by Kristensen et al. reports spatially resolved cellular expression profiles of a circular RNA (ciRS-7) in colon cancer. ciRS-7 had previously been proposed to act as a sponge for miR-7, a microRNA tumour suppressor that targets several oncogenic factors. The authors show that ciRS-7 is actually completely absent in cancer cells and instead expressed highly in stromal cells in the tumour microenvironment. This observation contradicts previous suggestions that ciRS-7 can impact levels of oncogenic factors in cancer cells through an miR-7 sponging mechanism. Previous studies correlating ciRS-7 levels with miR-7 targeted oncogenic factors are instead explained by differences in cancer to stromal cell ratios. The authors conclude that ciRS-7 is unlikely to have any driving role in colon cancer and adenocarcinoma generally. The findings have significant implications on studies linking circular RNAs with expression levels of microRNA targets in the context of the competitive endogenous RNA (ceRNA) sponge hypothesis. Critically, the study highlights the importance of spatial resolution of gene expression in forming mechanistic hypotheses.

The study is to this reviewer's knowledge the first of its kind and provides a valuable illustration of the limitations of examining expression profiles from heterogenous tissue samples. Circular RNAs are highly studied and increasingly recognised as potentially important biomarkers and potential drivers of disease or targets for treatment. Understanding their function and mechanism of action is critical and their often presumed role as miRNA sponges can and has led to spurious and potentially misleading investigations. This paper provides compelling evidence to challenge existing assumptions and provides an alternative model to interpret correlated (and inversely correlated) expression profiles.

One apparent limitation in the study is the absence of data on miR-7 expression. miR-7 is a key player in this story, and surprisingly the authors have not reported on its expression in the samples they studied. It would have been insightful to have seen its expression in stroma and cancer cells. This is especially relevant, as any sponging mechanism for circular RNAs relies on suitable stoichiometric ratios to act as a microRNA sponge in the first place (this is mentioned in the discussion as it is often not the case). It would have been informative for the reader to have seen the ratios of ciRS-7 : miR-7 and raised whether or not ciRS-7 occurred in sufficient quantities to appreciably impact miR-7 levels in any case.

The manuscript is generally well written and presented and the statistical analyses are appropriate. As a minor comment, the manuscript would benefit from removal of the "clearly" superlative, which is used a few times.

Reviewer #2 (Remarks to the Author):

In this manuscript entitled "The putative oncogene ciRS-7 is not expressed in colon cancer cells: implications for the ceRNA hypothesis", the authors spatially visualized the expression of circRNA ciRS-7 by RNA chromogenic in situ hybridization (CISH) assay in colon cancer samples. They surprisingly showed that ciRS-7 was exclusively expressed in stromal, but not in colon cancer, cells within tumors. In addition, the authors also tested correlations between ciRS-7 and miR-7-targeted genes, and found both positive and negative correlations between them. To further understand the positive correlation, the authors knocked out ciRS-7, but found that the expression of positively-correlated FOS gene was not significantly changed. These findings argued against the notion that ciRS-7 functions through ceRNA model, at least in colon cancer. Finally, the authors proposed a model to explain the expression correlation between any circRNAs, including ciRS-7, and miR-7-targeted genes. They concluded that circRNAs could be correlated with miR-7-targeted genes depending on whether they are co-expressed or mutually exclusive in cancer- and stromal

cells, but instead of simply explaining by the ceRNA model. Together, this study carefully examined the spatial expression of ciRS-7 circRNA in cancers, and demonstrated the complex regulation of ciRS-7 within cancer. These new findings are very useful to the circRNA field. There are some suggestions for the authors to consider for their revision.

1. Although the authors stated that “we believe that our data are not in conflict with the majority of the previous findings”, their findings were different to previous ones, at least for the spatial expression of ciRS-7 circRNA within tumors. These findings also argue against using ceRNA to simply explain the regulation of ciRS-7 with miR-7-targeted genes in colon cancer. Despite that more details are needed to address the underlined mechanism(s), these arguments are very important to the field to clarify previous misunderstanding/misinterpretation about the regulation of circRNAs in cancer. In this case, the authors may like to directly state something in their Discussion as: “we showed some different results/findings...”.

2. Some of their data are indeed different to previous reports. For example, to confirm the model (Figure 3), authors investigate the correlation between expression of ciRS-7 and miR-7-targeted genes. As shown in Figure 4, expression of ciRS-7 has negative correlation with that of EGFR, which is opposite with that in a previous research (Figure 5 in PMID: 28174233).

3. In support of their proposed new model (Figure 3), the authors selected more circRNAs (including circCCDC66, circFBXW7, circZKSCAN1 and circZNF91) without miR-7 target sites to analyze their correlation with miR-7-targeted genes (Figure 5). Can this comparison be extended to other genes that are not targeted by miR-7? How about to compare these spatially-expressed circRNAs with genes targeted by other types of cancer-related microRNAs? By analyzing this, the authors may further strengthen their model and conclusions. Obviously, the underlined mechanism is not clear, and it can be out of scope of the current study.

4. There are some minor mistakes. On page 4, author stated that “The T3 transcript of LINC00632 drives ciRS-7 expression in tumor stromal cells”, and the main evidence shown in the manuscript is that the expression of T3 correlated positively with that of ciRS-7, which maybe not enough to draw the conclusion of “The T3 transcript of LINC00632 drives ciRS-7 expression”.

At line 109, “Fig. 1f” should be “Fig. 1h”.

At line 110, “Fig. 1g,h” should be “Fig. 1f,g”.

Reviewer #3 (Remarks to the Author):

The manuscript by Kristensen et al studies the expression pattern and function of the circular RNA ciRS-7, with particular focus and critical examination of its potential role as a competing endogenous RNA (ceRNA). It has previously been suggested that ciRS-7, also known as CDR1as, is a ceRNA due to its large number of miR-7 binding sites (63) and potential differential expression in bulk tumor tissues. However, these previous largely correlative studies have failed to pinpoint to the physiological importance and precise mechanism of action of ciRS-7.

This manuscript makes the important observation that ciRS-7 is not expressed in cancer cells but appears to show signal in the tumor-infiltrating stromal compartment. This finding adds to a growing body of work which has revealed that many genes, including miRNAs, can show misleading depletion or enrichment of their expression in bulk cancer tissues compared to normal tissues due to varied levels of infiltration of stromal cells. While investigating this topic is timely and important, the present work seems premature and incomplete. To complete the story, the authors would need to perform revisions and further work, as outlined below.

1) Missing controls for CISH detection: The CISH signal is clearly specific to stromal compared to tumor/epithelial cells. However, there is no evidence that it's specific to ciRS-7. The probe could be detecting another stromal-specific gene. As a suggestion, the authors could use their knockout model and perform CISH detection in paraffin blocks prepared from WT and KO HEK cells to confirm circRNA specificity.

2) Detection of miR-7 is missing: This story would benefit tremendously from in situ detection of miR-7 in sections from the same tissues where ciRS-7 was detected.

3) The authors should examine the effects on all predicted miR-7 targets rather than focus on hand-picked miRNA targets: Rather than focusing on individual miR-7 target genes, proper computation analysis should be performed to examine the effects on all predicted miR-7 target genes (TargetScan) using cumulative frequency distribution plots. Genes with strong (8-mer) binding sites in their 3'UTRs versus weaker binding sites can also be examined.

4) Direct comparison of the expression patterns of stromal and cancer cells is flawed: When focusing on miR-7 target genes, direct comparison of their expression levels cannot be made between different cell types because such changes could, in principle, be due to cell type-specific differences of basal expression levels, miR-7 levels, abundance of other miR-7 targets, in addition to ciRS-7 status. The only way to examine the ceRNA hypothesis is to compare same cell type with and without miR-7 or ciRS-7 expression.

5) HEK293 are a poor model for studying a stromal-specific gene: Given the evidence that ciRS-7 is preferentially expressed in stromal and not in epithelial cells, the model system they have chosen is flawed. KO should be performed in a stromal cell type; epithelial KO could be a possible negative control.

In summary, while the reviewer agrees with the general conclusion of this study and agrees that it makes an important conceptual contribution, given the previous publications of flawed studies. However, the reviewer encourages the researchers to extend their work to strengthen their conclusions.

Reviewer #1 (Remarks to the Author):

The manuscript by Kristensen et al. reports spatially resolved cellular expression profiles of a circular RNA (ciRS-7) in colon cancer. ciRS-7 had previously been proposed to act as a sponge for miR-7, a microRNA tumour suppressor that targets several oncogenic factors. The authors show that ciRS-7 is actually completely absent in cancer cells and instead expressed highly in stromal cells in the tumour microenvironment. This observation contradicts previous suggestions that ciRS-7 can impact levels of oncogenic factors in cancer cells through an miR-7 sponging mechanism. Previous studies correlating ciRS-7 levels with miR-7 targeted oncogenic factors are instead explained by differences in cancer to stromal cell ratios. The authors conclude that ciRS-7 is unlikely to have any driving role in colon cancer and adenocarcinoma generally. The findings have significant implications on studies linking circular RNAs with expression levels of microRNA targets in the context of the competitive endogenous RNA (ceRNA) sponge hypothesis. Critically, the study highlights the importance of spatial resolution of gene expression in forming mechanistic hypotheses.

We agree with the reviewer that our findings have significant implications for future studies in the context of the ceRNA sponge hypothesis and that our study highlights the importance of spatial resolution of gene expression in forming mechanistic hypotheses.

The study is to this reviewer's knowledge the first of its kind and provides a valuable illustration of the limitations of examining expression profiles from heterogenous tissue samples. Circular RNAs are highly studied and increasingly recognised as potentially important biomarkers and potential drivers of disease or targets for treatment. Understanding their function and mechanism of action is critical and their often presumed role as miRNA sponges can and has led to spurious and potentially misleading investigations. This paper provides compelling evidence to challenge existing assumptions and provides an alternative model to interpret correlated (and inversely correlated) expression profiles.

We thank the reviewer for the kind words and agree that our paper provides compelling evidence to challenge existing assumptions.

One apparent limitation in the study is the absence of data on miR-7 expression. miR-7 is a key player in this story, and surprisingly the authors have not reported on its expression in the samples they studied. It would have been insightful to have seen its expression in stroma and cancer cells. This is especially relevant, as any sponging mechanism for circular RNAs relies on suitable stoichiometric ratios to act as a microRNA sponge in the first place (this is mentioned in the discussion as it is often not the case). It would have been informative for the reader to have seen the ratios of ciRS-7 : miR-7 and raised whether or not ciRS-7 occurred in sufficient quantities to appreciably impact miR-7 levels in any case.

We agree with the reviewer that it is important to investigate miR-7 expression in stroma and cancer cells within the tumor microenvironment. We did this by profiling the expression of 799 miRNAs, including miR-7, using NanoString nCounter technology within laser capture microdissected (LCMD) patient samples (see Fig. 2m). The following was written in the Result section:

Moreover, we profiled the expression of 799 miRNAs in the laser capture microdissected fractions of cancer- and stromal cells, and found that miR-7 was only expressed in the cancer cells and not detectable in the

stromal cells (Fig. 2m and Supplementary Table 1). Again, arguing against explaining the observed correlations using the ceRNA hypothesis.

We also commented on these results in the discussion section:

We found that miR-7 was not expressed at detectable levels in ciRS-7 expressing stromal cells, also arguing against explaining the overserved correlations by the ceRNA hypothesis.

We have now further addressed this by performing *in situ* analysis for miR-7 using a novel miRNAscope technology. These new data are in line with what we observed using NanoString nCounter analysis on LCMD samples and are presented as Supplementary Figure 6 in the new version of our manuscript.

The manuscript is generally well written and presented and the statistical analyses are appropriate. As a minor comment, the manuscript would benefit from removal of the “clearly” superlative, which is used a few times.

We agree and have now removed “clearly” from the following sentences in the discussion section:

we ~~clearly~~ demonstrate that this circRNA is not present in colon cancer cells and several other human adenocarcinomas.

Thirdly, we have ~~clearly~~ demonstrated that the expression of circRNAs correlate with the expression of miR-7 target genes independent of whether the circRNAs harbor miR-7 binding sites.

However, differences in cancer-to-stromal cell ratios among tumor specimens ~~clearly~~ can confound the analyses or even result in correlations that may be wrongly interpreted as evidence of ceRNA functions in cancer.

Reviewer #2 (Remarks to the Author):

In this manuscript entitled “The putative oncogene ciRS-7 is not expressed in colon cancer cells: implications for the ceRNA hypothesis”, the authors spatially visualized the expression of circRNA ciRS-7 by RNA chromogenic in situ hybridization (CISH) assay in colon cancer samples. They surprisingly showed that ciRS-7 was exclusively expressed in stromal, but not in colon cancer, cells within tumors. In addition, the authors also tested correlations between ciRS-7 and miR-7-targeted genes, and found both positive and negative correlations between them. To further understand the positive correlation, the authors knocked out ciRS-7, but found that the expression of positively-correlated FOS gene was not significantly changed. These findings argued against the notion that ciRS-7 functions through ceRNA model, at least in colon cancer. Finally, the authors proposed a model to explain the expression correlation between any circRNAs, including ciRS-7, and miR-7-targeted genes. They concluded that circRNAs could be correlated with miR-7-targeted genes depending on whether they are co-expressed or mutually exclusive in cancer- and stromal cells, but instead of simply explaining by the ceRNA model. Together, this study carefully examined the spatial expression of ciRS-7 circRNA in cancers, and demonstrated the complex regulation of ciRS-7 within cancer. These new findings are very useful to the circRNA field. There are some suggestions for the authors to consider for their revision.

We thank the reviewer for all the kind words and agree that our new findings are very useful to the circRNA field.

1. Although the authors stated that “we believe that our data are not in conflict with the majority of the previous findings”, their findings were different to previous ones, at least for the spatial expression of ciRS-7 circRNA within tumors. These findings also argue against using ceRNA to simply explain the regulation of ciRS-7 with miR-7-targeted genes in colon cancer. Despite that more details are needed to address the underlined mechanism(s), these arguments are very important to the field to clarify previous misunderstanding/misinterpretation about the regulation of circRNAs in cancer. In this case, the authors may like to directly state something in their Discussion as: “we showed some different results/findings...”.

We thank the reviewer for this comment, as it highlights that we may not have conveyed our message clearly enough in the first version of the manuscript. Thus, we think that the reviewer might have misunderstood our intention with stating that our data are not in conflict with the majority of the previous findings. What we were actually trying to convey in the discussion is that most of our data are not in conflict with previous data, but rather it is the interpretations of the data, which are in conflict. This is mainly due to the lack of spatial expression analyses of ciRS-7 in previous studies. We have tried to clarify this by modifying the following sentence in the discussion section:

“However, we believe that our data are not in conflict with the majority of the previous findings.”

So that it now reads:

“However, we believe that our data are not in conflict with the majority of the ~~previous findings~~-published data, but rather it is the interpretations of the data that are in conflict.”

2. Some of their data are indeed different to previous reports. For example, to confirm the model (Figure 3), authors investigate the correlation between expression of ciRS-7 and miR-7-targeted genes. As shown in Figure 4, expression of ciRS-7 has negative correlation with that of EGFR, which is opposite with that in a previous research (Figure 5 in PMID: 28174233).

The reviewer is correct that ciRS-7 was positively correlated with EGFR in the mentioned article (PMID: 28174233). Therefore, we have added the following sentence to the discussion section:

“Finally, while a previous study found a positive correlation between ciRS-7 and EGFR expression¹⁹, we observed a negative correlation between the two.”

3. In support of their proposed new model (Figure 3), the authors selected more circRNAs (including circCCDC66, circFBXW7, circZKSCAN1 and circZNF91) without miR-7 target sites to analyze their correlation with miR-7-targeted genes (Figure 5). Can this comparison be extended to other genes that are not targeted by miR-7? How about to compare these spatially-expressed circRNAs with genes targeted by other types of cancer-related microRNAs? By analyzing this, the authors may further strengthen their model and conclusions. Obviously, the underlined mechanism is not clear, and it can be out of scope of the current study.

In our first version of the manuscript, we provided evidence that *FOS* (the gene that showed the strongest positive correlation with ciRS-7) expression levels do not respond to altered miR-7 expression in a luciferase reporter setup (Fig. 2k,l). These results were confirmed using a dual-color fluorescent reporter (ppC) that allows single-cell measurements (Supplementary Fig. 4), and corroborated by the fact that human *FOS*, in contrast to the mouse version of *FOS*, does not contain any 7-mer or 8-mer target sites for miR-7 according to TargetScan. In addition, we found no change in *FOS* expression upon CRISPR/Cas9 based knockout of ciRS-7 (Fig. 2g-l). Therefore, we believe that we have provided evidence that our findings can be extended to other genes that are not targeted by miR-7. However, we agree with the reviewer that our study raises a number of interesting research questions, which should be addressed in the future, including how our findings may be extended to other potential ceRNAs and the corresponding miRNAs and target genes, but in line with the reviewer, we think this is outside the scope of the current study.

4. There are some minor mistakes. On page 4, author stated that “The T3 transcript of LINC00632 drives ciRS-7 expression in tumor stromal cells”, and the main evidence shown in the manuscript is that the expression of T3 correlated positively with that of ciRS-7, which maybe not enough to draw the conclusion of “The T3 transcript of LINC00632 drives ciRS-7 expression”.

At line 109, “Fig. 1f” should be “Fig. 1h”.

At line 110, “Fig. 1g,h” should be “Fig. 1f,g”.

We thank the reviewer for pointing out these mistakes, which have been corrected in the revised version of the manuscript.

We have also changed the following sentence:

The T3 transcript of LINC00632 drives ciRS-7 expression

To:

The T3 transcript of LINC00632 is likely to drive ciRS-7 expression in tumor stromal cells

Reviewer #3 (Remarks to the Author):

The manuscript by Kristensen et al studies the expression pattern and function of the circular RNA ciRS-7, with particular focus and critical examination of its potential role as a competing endogenous RNA (ceRNA). It has previously been suggested that ciRS-7, also known as CDR1as, is a ceRNA due to its large number of miR-7 binding sites (63) and potential differential expression in bulk tumor tissues. However, these previous largely correlative studies have failed to pinpoint to the physiological importance and precise mechanism of action of ciRS-7.

This manuscript makes the important observation that ciRS-7 is not expressed in cancer cells but appears to show signal in the tumor-infiltrating stromal compartment. This finding adds to a growing body of work which has revealed that many genes, including miRNAs, can show misleading depletion or enrichment of their expression in bulk cancer tissues compared to normal tissues due to varied levels of infiltration of stromal cells. While investigating this topic is timely and important, the present work seems premature and

incomplete. To complete the story, the authors would need to perform revisions and further work, as outlined below.

We thank the reviewer for the kind words and agree that this topic is timely and important. We have now carried out more experiments to make the work complete as detailed below and in response to the other reviewers comments.

1) Missing controls for CISH detection: The CISH signal is clearly specific to stromal compared to tumor/epithelial cells. However, there is no evidence that it's specific to ciRS-7. The probe could be detecting another stromal-specific gene. As a suggestion, the authors could use their knockout model and perform CISH detection in paraffin blocks prepared from WT and KO HEK cells to confirm circRNA specificity.

We thank the reviewer for this comment, but would like to state that the CISH assay is based on target RNA-specific oligonucleotide Z probes, which are hybridized in pairs (ZZ). This ensures a high specificity for the target, since both probes need to hybridize adjacent to each other before a signal is created. Because it is highly unlikely that two nonspecific hybridization events will juxtapose a pair of target probes along an off-target RNA molecule (PMID: 22166544) it is highly unlikely that our assay target another stromal-specific gene. The probes are commercially available and the assay has previously been published (PubMed PMID: 31775754). However, we have validated the CISH data using laser capture microdissection of the colon cancer tissues to study cancer cells and stromal cells separately using NanoString nCounter technology. The NanoString probes for ciRS-7 are in a different location compared to the CISH probes, and we have previously validated that this assay is specific for circular RNA by treatment of total RNA samples with RNase R (an enzyme that only degrades linear RNA). In addition, we validated the NanoString data using RNA-seq and RT-qPCR (PMID: 30087459). Moreover, we find that the ciRS-7 signal is specific to the activated stroma within the tumor microenvironment (negative in uninvolved stroma) and completely absent in normal colon epithelial cells (Fig. 1a,b and Supplementary Fig. 1). Finally, we also observed that cancer cells of neuroendocrine carcinomas express ciRS-7. This also applied to malignant melanoma (Supplementary Fig. 8), which arises from the neural crest-derived melanocytes, whereas several adenocarcinomas showed similar spatial expression patterns of ciRS-7 as in colon cancer (negative in the cancer cells and highly expressed in stromal cells) (Fig. 6). Since ciRS-7 is highly expressed in neuronal tissues (PMID: 29887379, PMID: 28798046, PMID: 23446348, PMID: 23446346), we do not believe that these observations would not have been made with a CISH assay that is detecting a stromal-specific gene.

Nevertheless, we have now provided further evidence that the CISH signal is specific to ciRS-7 by performing the assay on paraffin blocks prepared from wild-type and knock-out HEK cells as suggested by the reviewer. These data have been included in the new version of the manuscript (See Supplementary Fig. 3), and the following sentence was added in the result section:

Importantly, we confirmed that our CISH assay is specific for ciRS-7 by analyzing ciRS-7 knock-out and wild-type cells (Supplementary Fig. 3).

2) Detection of miR-7 is missing: This story would benefit tremendously from in situ detection of miR-7 in sections from the same tissues where ciRS-7 was detected.

We agree with the reviewer that it is important to investigate miR-7 expression in the same tissues where ciRS-7 was detected. We did this by profiling the expression of 799 miRNAs, including miR-7, using NanoString nCounter technology within laser capture microdissected (LCMD) patient samples (i.e. in the stromal cell and cancer cell fractions within the tumor microenvironment) (Fig. 2m). The following was written in the Result section:

Moreover, we profiled the expression of 799 miRNAs in the laser capture microdissected fractions of cancer- and stromal cells, and found that miR-7 was only expressed in the cancer cells and not detectable in the stromal cells (Fig. 2m and Supplementary Table 1). Again, arguing against explaining the observed correlations using the ceRNA hypothesis.

In addition, we have now confirmed these results by performing *in situ* analysis for miR-7 in sections from the same tissues where ciRS-7 was detected, as suggested by the reviewer, using a novel miRNAscope technology. These new data are in line with the NanoString data on LCMD samples and are presented as Supplementary Figure 6 in the new version of our manuscript.

3) The authors should examine the effects on all predicted miR-7 targets rather than focus on hand-picked miRNA targets: Rather than focusing on individual miR-7 target genes, proper computation analysis should be performed to examine the effects on all predicted miR-7 target genes (TargetScan) using cumulative frequency distribution plots. Genes with strong (8-mer) binding sites in their 3'UTRs versus weaker binding sites can also be examined.

The miR-7 target genes were not hand-picked, but selected *a priori* based on their relevance in cancer and experimental evidence being present that the genes are *bona fide* miR-7 target genes. Though we agree with the reviewer that the suggested analyses would be of interest, it is not possible to do this analysis, as we did not perform RNA-seq in this study, which is based on formalin-fixed paraffin-embedded (FFPE) tissues. We have tried to perform total RNA-sequencing using RNA extracted from FFPE tissues without success (i.e. almost no backsplicing junction-specific sequencing reads were present in the data). Therefore, it is not meaningful to include these data in the manuscript.

4) Direct comparison of the expression patterns of stromal and cancer cells is flawed: When focusing on miR-7 target genes, direct comparison of their expression levels cannot be made between different cell types because such changes could, in principle, be due to cell type-specific differences of basal expression levels, miR-7 levels, abundance of other miR-7 targets, in addition to ciRS-7 status. The only way to examine the ceRNA hypothesis is to compare same cell type with and without miR-7 or ciRS-7 expression.

We agree with the reviewer. This is actually one of the main messages of our manuscript. Indeed, we show that differences in cancer-to-stromal cell ratios among tumor specimens may confound the analyses and result in correlations that are often wrongly interpreted as evidence of ceRNA functions in cancer.

5) HEK293 are a poor model for studying a stromal-specific gene: Given the evidence that ciRS-7 is preferentially expressed in stromal and not in epithelial cells, the model system they have chosen is flawed. KO should be performed in a stromal cell type; epithelial KO could be a possible negative control.

We agree with the reviewer that the HEK293T cells would have been a poor model if these cells did not express ciRS-7 and FOS. However, we chose the HEK293T cells, as these express both ciRS-7 and FOS at relatively high levels.

In summary, while the reviewer agrees with the general conclusion of this study and agrees that it makes an important conceptual contribution, given the previous publications of flawed studies. However, the reviewer encourages the researchers to extend their work to strengthen their conclusions.

We thank the reviewer for all the constructive comments and are happy that he/she agrees with the general conclusion of this study and agrees that it makes an important conceptual contribution to the field.

REVIEWERS' COMMENTS:

Reviewer #1 (Remarks to the Author):

I am satisfied that the authors have addressed my key concern regarding the manuscript through the addition of miR-7 in situ data. As stated in my original review, the findings are novel, the data presented are convincing and the overall claims are of broad interest to the community and wider field.

Marcel E. Dinger, UNSW Sydney.

Reviewer #2 (Remarks to the Author):

This manuscript has been revised accordingly, and is suitable for publication.

Reviewer #3 (Remarks to the Author):

The authors have addressed all the concerns that I had. In particular, the edited text conveys the main points more strongly and more accurately than before. Recommend publication.